# Novel Insight into the Effects of CpxR on *Salmonella enteritidis* Cells during the Chlorhexidine Treatment and Non-Stressful Growing Conditions

**DOI:** 10.3390/ijms22168938

**Published:** 2021-08-19

**Authors:** Xiaoying Liu, Misara Omar, Kakambi V. Nagaraja, Sagar M. Goyal, Sinisa Vidovic

**Affiliations:** 1Department of Veterinary and Biomedical Sciences, University of Minnesota, Saint Paul, MN 55108, USA; liux2725@umn.edu (X.L.); omarx119@umn.edu (M.O.); nagar001@umn.edu (K.V.N.); 2Veterinary Diagnostic Laboratory, Department of Veterinary Population Medicine, University of Minnesota, Saint Paul, MN 55108, USA; goyal001@umn.edu; 3Food Safety and Preservation Team, The New Zealand Institute for Plant and Food Research Limited, Auckland 1025, New Zealand

**Keywords:** *Salmonella*, CpxR regulator, chlorhexidine, antimicrobial resistance, proteomics

## Abstract

The development and spread of antibiotics and biocides resistance is a significant global challenge. To find a solution for this emerging problem, the discovery of novel bacterial cellular targets and the critical pathways associated with antimicrobial resistance is needed. In the present study, we investigated the role of the two most critical envelope stress response regulators, RpoE and CpxR, on the physiology and susceptibility of growing *Salmonella* *enterica* serovar *enteritidis* cells using the polycationic antimicrobial agent, chlorhexidine (CHX). It was shown that deletion of the *cpxR* gene significantly increased the susceptibility of this organism, whereas deletion of the *rpoE* gene had no effect on the pathogen’s susceptibility to this antiseptic. It has been shown that a lack of the CpxR regulator induces multifaceted stress responses not only in the envelope but also in the cytosol, further affecting the key biomolecules, including DNA, RNA, and proteins. We showed that alterations in cellular trafficking and most of the stress responses are associated with a dysfunctional CpxR regulator during exponential growth phase, indicating that these physiological changes are intrinsically associated with the lack of the CpxR regulator. In contrast, induction of type II toxin-antitoxin systems and decrease of abundances of enzymes and proteins associated with the recycling of muropeptides and resistance to polymixin and cationic antimicrobial peptides were specific responses of the ∆*cpxR* mutant to the CHX treatment. Overall, our study provides insight into the effects of CpxR on the physiology of *S*. Enteritidis cells during the exponential growth phase and CHX treatment, which may point to potential cellular targets for the development of an effective antimicrobial agent.

## 1. Introduction

The development and spread of antibiotics and biocides resistance among various human pathogens poses one of the greatest challenges in 21st century human medicine [1,2,3,4]. The World Health Organization, considering experts opinion and existing data, created a priority list of antibiotic-resistant bacteria. This list contained three priority tiers, including critical, high, and medium [5]. According to this list, Gram-negative pathogens were exclusively classified in the critical group, indicating that Gram-negative bacteria pose significant threats. One of the reasons why these organisms pose such a threat is associated with their unique structural features, outer membrane, and periplasmic space (a part of the Gram-negative envelope), which serve as a selectively permeable barrier and therefore increase the ability of these bacteria to resist antimicrobial treatments.

The alternative sigma factor, RpoE (σ^E^), and two component signal transduction system, CpxRA, play crucial roles in coping with the envelope or extracytoplasmic stress responses [6,7]. The RpoE regulator controls a conserved group of genes across various microbial species and a variable group of genes depending on the lifestyle of each microbial species [8]. The conserved regulon of the RpoE is associated with the biosynthesis, maintenance, and repair of lipopolysaccharides (LPS) and porins, essential elements for cellular trafficking in Gram-negative bacteria [9]. Conversely, the Cpx regulon is involved in various metabolic processes associated with the inner membrane, including cellular trafficking [10], cellular respiration, [11] and maintenance as well as repair of the inner membrane [12]. Besides the inner membrane, the Cpx regulator is involved in the biogenesis of bacterial appendages [13]. Moreover, it has been reported that Cpx controls the expression of the genes encoding proteins and enzymes associated with phospholipid metabolism, assembly, and maintenance of outer membrane porins [14,15]. Both envelope stress response regulators are involved in crucial processes of survival and pathogenicity [16,17]. The activation of the RpoE regulon is initiated by the presence of misfolded outer membrane proteins and off-pathway components in LPS transport and assembly [8,18], while the Cpx response mechanism is triggered with the misfolded periplasmic proteins [19]. Although these two envelope stress regulators share multiple connections for responses, it has been shown that the Cpx can act antagonistically to RpoE, mainly by repressing the synthesis of outer membrane ß-barrel proteins [20] and chaperones [21] under control of RpoE. Both envelope stress response regulators play a role in the susceptibility of Gram-negative bacteria to different classes of antibiotics [22,23,24,25].

In the current study, we investigated the role of both RpoE and CpxR, regulators on the susceptibility of growing *Salmonella enterica* serovar Enteritidis cells to the polycationic antimicrobial agent, chlorhexidine (CHX). The mechanism of action of CHX is associated with the interaction of positively charged CHX molecules with the negatively charged surface of a microbial cell. As a result of this interaction, the cell membrane tends to disorganize, leading to a “leaky cell” phenotype. To study the roles of major extracytoplasmic stress response regulators in antimicrobial susceptibility, we used an antimicrobial agent that primarily acts on the cell membrane and subsequently triggers the extracytoplasmic stress response. Our study showed that deletion of the *cpxR* gene significantly increased the susceptibility of this organism, whereas deletion of the *rpoE* gene did not affect the susceptibility of pathogen to this antiseptic. We showed that the lack of a functional CpxR regulator induces multifaceted stress responses, not only in the envelope but also in the cytosol, further affecting the key biomolecules, including DNA, RNA, and proteins. It was shown that alterations in cellular trafficking and most of the stress responses are intrinsically associated with the lack of the CpxR regulator. Induction of type II toxin-antitoxin systems and decrease of abundances of enzymes and proteins associated with the recycling of muropeptides and resistance to polymixin and cationic antimicrobial peptides were specific responses of the ∆*cpxR* mutant to the CHX treatment.

## 2. Results

### 2.1. Growth Kinetics

To determine whether there was any effect of *cpxR* and *rpoE* mutations on the growth rate of *S.* Enteritidis, we carried out an assay measuring the growth rates of Δ*cpxR* and Δ*rpoE* mutants and compared them with the growth rates of the parental strain (wild-type strain). There was no difference in the growth rates between the wild-type and the Δ*cpxR* mutant, except for the last measurement (Figure 1), suggesting that virtually the *cpxR* deletion does not affect the growth rate of *S.* Enteritidis.

It is worth noting that the significant difference in biomass between the Δ*cpxR* mutant and the wild-type, which occurred after 10 h of incubation, is associated with the organisms’ ability to resist starvation rather than with their growth rates, as both organisms at that time point entered the stationary growth phase. Opposite to the Δ*cpxR* mutant, the Δ*rpoE* mutant strain showed a statistically significant difference (*p* < 0.05) after a 4 h incubation compared with those of the wild-type and the Δ*cpxR* mutant (Figure 1). This trend of a significantly lower growth rate (*p* < 0.05) of the Δ*rpoE* mutant lasted until the beginning of the stationary growth phase when a natural growth arrest of the *S.* Enteritidis culture occurred (Figure 1), further indicating that the *rpoE* deletion imposes a growth deficiency in this organism.

### 2.2. Determination of the Minimum Inhibitory Concentration (MIC) of CHX

The objective of this assay was to find out the MIC of CHX (Figure 2A) to the wild-type and to employ this MIC in subsequent experiments to examine the effects of this antiseptic on the Δ*rpoE* and Δ*cpxR* mutants.

Initially, the antimicrobial susceptibility of the wild-type was tested using a wide range of CHX concentrations (25, 20, 15, 10, 5, and 1 µg/mL). Analysis of these data revealed that the concentrations of CHX greater than 5 µg/mL imposed a bactericidal effect, whereas 5 µg/mL exhibited a bacteriostatic effect (Figure 2B). The concentration of 1 µg/mL had no antimicrobial effect on the wild-type (Figure 2B). To determine the MIC of CHX, concentrations lower than 5 and greater than 1 µg/mL were selected for the next round of screenings. After three biological replications, a concentration of 3 µg/mL showed a persistent bacteriostatic effect, while a concentration of 2 µg/mL could not inhibit the growth of the wild-type (Figure 2C). This assay demonstrated that 3 µg/mL of CHX represents the minimum inhibitory concentration.

### 2.3. Effects of the Drug Treatment on the Viability of the ΔrpoE and ΔcpxR Mutants

To evaluate the effects of CHX on the Δ*rpoE* and Δ*cpxR* mutants and to reveal the role of these two extracytoplasmic stress response regulators in antimicrobial susceptibility of *S.* Enteritidis to a polycationic agent, we carried out a series of CHX treatment assays. Deletion of the *rpoE* gene had no effect on the antimicrobial susceptibility of *S.* Enteritidis during the exponential growth phase, as no significant difference in the cell viability was observed between the wild-type and Δ*rpoE* mutant strains (Figure 3A).

In contrast to the Δ*rpoE* mutant, deletion of the *cpxR* gene caused a significant difference (*p* < 0.05) in the cell viability between the wild-type and Δ*cpxR* mutant (Figure 3A). The average value of 8.741 on the log_10_ scale for the Δ*cpxR* mutant was significantly lower than the average value of 9.007 on the log_10_ scale for the wild-type throughout the CHX treatment (Figure 3A). As the Lambda Red recombineering, a molecular cloning approach based on homologous recombination, leaves a scar in the genome, usually in a form of a DNA insertion, this DNA manipulation approach can subsequently cause a frameshift mutation, further affecting the existing genes in the polycistronic operon of the targeted gene. To rule out any polar effect associated with the increased susceptibility of the Δ*cpxR* mutant to CHX, the isogenic *cpxR* mutant was complemented with a functional *cpxR* gene, followed by a CHX treatment assay. The Δ*cpxR* mutant complemented with the functional *cpxR* gene was able to mainly restore its resistance to CHX similarly to that of its parental strain (Figure 3B). This indicates that the increased CHX susceptibility of the Δ*cpxR* mutant is due to the lack of *cpxR* gene and not to the side effects of *cpxR* deletion.

### 2.4. The cpxR Mutation Does Not Cause the Major Perturbation of the Cellular Envelope during the Initial CHX Treatment

The transmission electron microscopy (TEM) analyses (Figure 4A–H) was performed to determine the effects of the CHX treatments and/or *cpxR* deletion on the envelope integrity of the *S.* Enteritidis strain.

The samples for the TEM analyses were taken after 60 min of CHX treatment when the first significant difference (*p* < 0.05) in the CHX susceptibility between the Δ*cpxR* mutant and its parental strain was observed. Comparing the wild-type untreated (Figure 4A,B) with the same organism treated with CHX (Figure 4C,D), it can be observed that the cellular turgor and outer membrane integrity were the same, suggesting that the MIC of CHX does not compromise cell structure integrity of the wild-type. Next, using the wild-type untreated (Figure 4A,B) as the reference micrographs and comparing them with the micrographs of the untreated Δ*cpxR* mutant (Figure 4E,F) and the treated Δ*cpxR* mutant (Figure 4G,H), it can be observed that neither the *cpxR* deletion nor a combination of *cpxR* deletion and CHX treatment affected the cell structure integrity. This indicates that the initial lethality of the Δ*cpxR* mutant during the CHX treatment was not associated with major structural perturbation of the bacterial envelope, but rather via less invasive changes of the cellular envelope, primarily outer membrane and secondary periplasm and inner (plasma) membrane.

### 2.5. Effect of the CpxR Mutation on the Proteome of S. enteritidis during the Exponential Growth Phase

To reveal the effect of the *cpxR* mutation on the overall physiology of *S*. Enteritidis, we compared proteomes of the wild-type and the Δ*cpxR* mutant during the exponential growth phase. In total, out of 2327 proteins identified by Isobaric Tag for Relative and Absolute Quantification (iTRAQ) in conjunction with High Performance Liquid Chromatography (HPLC) (Figure 5A), 596 proteins showed significant differences based on the permutation test and Benjamini-Hochberg method (*p* < 0.023). The altered proteome of the Δ*cpxR* mutant contained 347 proteins with increased abundances and 249 proteins with decreased abundances compared with the wild-type (Figure 5A). Appendix A presents the full list of significantly altered proteins, along with their gene names, functions, accession numbers, molecular weights, a permutation test, and Benjamini-Hochberg *p*-values as well as log_2_ fold and fold changes.

The notable change imposed with the *cpxR* deletion was the alteration of enzymes, specifically enzymes/proteins associated with the bacterial envelope, either with membrane biogenesis or membrane transport (Figure 5B). The *cpxR* deletion caused an increase in abundance of enzymes involved in biosynthetic processes of the peptidoglycan layer, MurL, MurB, LdcA, and YcfS, as well as enzymes involved in ligation of amino acids to UDP-*N*-acetylmuramic acid, MurE, and Ddl (Table 1).

Besides enzymes involved in the biosynthesis of the peptidoglycan, a group of enzymes associated with peptidoglycan-recycling processes, AmpD and MltC, and the lipoprotein Lpp that links the inner leaflet of the lipopolysaccharides (LPS) to the peptidoglycan layer, showed increased abundance (Table 1). Another group of enzymes, AmiA, Alr, DacD, AmpH and LtdT, involved in peptidoglycan maturation and remodelling, showed a decreased abundance in the *cpxR* mutant (Table 1). An altered proteome in the *cpxR* mutant was associated with LPS biogenesis, including enzymes that are involved in the biosynthesis of all three components of LPS (lipid A, the core of LPS and O-antigen). The mutant showed an increased abundance of LpxA, LpxL, LpxC, LpxO, EptA, ArnA, enzymes associated with the biosynthesis of lipid A. Moreover, the mutant showed an increased abundance of glycosyltransferases, including CobK, RffM, and RfbN, which are critical in the biosynthesis of the core LPS and *O*-antigen, respectively (Table 1).

Along with enzymes/proteins associated with the biogenesis of the bacterial envelope, the numerous membrane transport systems showed a significant alteration in the Δ*cpxR* background compared with that of the wild-type. Most commonly, it was observed as an increased abundance of membrane transporters associated with the uptake of amino acids and peptides. The significant increase in abundance showed the transporters are linked with the uptake of glycine/proline (ProV), methionine (MetN), arginine (ArtJ), glutamate/aspartate (GltL), histidine (HisP), putrescine (PotF), cysteine/glutathione (CydC), and peptide (SapA). A small group of amino acid transporters, including branched-chain amino acid (LivK) and l-asparagine (AnsP), showed decreased abundances (Appendix A). There was also an increased abundance of the TonB/ExbB complex (Appendix A), a system linked with the uptake of large molecules such as iron-siderophore and vitamin B complexes. Iron ABC transporter (FetA), ferrous iron transporter (FeoB), and biotin (BisC) showed an increased abundance (Appendix A). Importantly, the Δ*cpxR* mutant showed an increased abundance of membrane transporters crucial for membrane integrity (TolQ, LptF, TamA, LolD, LptG) as well as transporters for metal (ZnuC, ModF), cell division (FtsE), sugar uptake (GntT, MglAB) and multidrug efflux (AdeB) (Appendix A). The membrane transporters associated with the tricarboxylic acid cycle (TCA), including TctC, GatB, Crr, PtsG, SrlE, UgpC, and GlpT, showed a significant reduction of abundance (Appendix A). Interestingly, the mutant exhibited a decrease in abundance of the major porin, OmpC, and another two porins, OmpN and Tsx, showed the same trend (Appendix A). Besides the TSA transporters and porins, the mutant showed a decrease in abundance of transporters involved in metal uptake (ModC, CorA, FeoA, FepC), phosphate uptake (PstB), nitrate uptake (NarK), maintenance of membrane lipid (MlaB), electron transport (DmsA, PhsB), and efflux (EmrD, AcrE) (Appendix A).

The *cpxR* deletion triggered alterations of stress response proteins associated with DNA repair (Figure 5C), folding of non-native polypeptides and proteolysis of irreversible aggregated proteins (Figure 5D) as well as degradation of RNA (Figure 5E). The Δ*cpxR* mutant exhibited an increased abundance of DNA repair proteins specifically associated with mismatch repair (LigA, HolAC, DnaQ, Ssb, MutLS, UvrD), nucleotide excision repair (LigA, Mfd, UvrD, NrdE), base excision repair (XthA, Nfo, LigA), SOS response (LexA), recombinational DNA repair (RdgC, RecB, DinG, RadA) and DNA replication (NrdE, RelB). Among this group of proteins, only DNA polymerase III subunit alpha (DnaG) and DNA polymerase III subunit theta (HolE) exhibited a decreased abundance in the Δ*cpxR* background. The Δ*cpxR* mutant manifested an increased abundance of various families of heat shock proteins (HSPs), including HSP 100 (ClpAB, HslU), HSP 60 (GroL), and HSP 40 (DnaJ, DjlA). Along with HSPs, the major oxidative stress response proteins, thioredoxin (TrxC), glutaredoxin 1 (GrxA), glutaredoxin 3 (GrxA), disulphide reductase (DipZ) and the key periplasmic chaperones (DsbA, DegP) showed the same trend. One HSP 70 (HscC) and a few chaperones involved in the biogenesis of type 1 fimbriae (FimC), maturation of trimethylamine N-oxide reductase TorA (TorD) and biosynthesis of (NiFe) hydrogenases (SlyD) showed decreased values compared with the wild-type. The entire altered proteome associated with RNA degradation showed increased abundance in the Δ*cpxR* mutant during the exponential growth phase (Figure 5E).

### 2.6. Response of the ΔcpxR Mutant to the Minimum Inhibitory Concentration of CHX

The effect of the *cpxR* deletion on *S.* Enteritidis susceptibility to CHX was examined using comparative proteomics of the wild-type and the Δ*cpxR* mutant with the MIC CHX-treatments. In total, 156 proteins exhibited significant shifts in abundance (≥1.5-fold up- or downregulations), with the significant differences based on the permutation test and Benjamini-Hochberg method (*p* < 0.023) and high reproducibility. Out of 156 proteins, 85 showed increased abundance while 71 proteins exhibited decreased abundance in the Δ*cpxR* background during the CHX treatment. The altered proteins were most commonly associated with central metabolic pathways (*n* = 54; 34.6%), following proteins of unknown function (*n* = 23; 14.7%), stress response proteins (*n* = 21; 13.4%), receptors and transporters (*n* = 19; 12.2%), transcription, replication, recombination and DNA maintenance (*n* = 19, 12.2%), virulence factors (*n* = 8, 5.1%), RNA processing (*n* = 5, 3.2%), ribosome-associated proteins (*n* = 4, 2.5%), motility (*n* = 2, 1.3%), and plasmid mobilization (*n* = 1, 0.6%) (Appendix A).

The most notable change imposed by the *cpxR* mutation was the alteration of proteins of the central metabolic pathways that are directly associated, functionally or structurally, with the envelope of *S*. Enteritidis. A great majority (86.6%) of these proteins showed a decreased abundance in the ∆*cpxR* background compared with that of the wild-type (Table 2). An identified group of downregulated proteins was involved in recycling of muropeptides during cell elongation and/or cell division (membrane-bound lytic murein transglycosylase (EmtA)), assembly of outer membrane proteins (AsmA family protein (AsmA), fructoselysine-6-*P*-deglycase (FrlB)), and resistance to polymyxin and cationic antimicrobial peptides (ArnC).

Furthermore, this group of proteins included the integral components of the membrane (PepE, FdxH, HemX, NfsB, WP_065618791.1, WP_000750393.1, WP_000748128.1, WP_001095011.1, and WP_001240360.1). Only two proteins involved in the biosynthesis of *O*-antigen (dTDP-glucose 4,6-dehydratase [RfbB]) and the alpha-glucan catabolic process (periplasmic alpha-amylase [MalS]) showed increased abundance in the ∆*cpxR* mutant (Table 2). Besides proteins associated with the envelope, there was decreased abundance in the ∆*cpxR* mutant of various dehydrogenases, including 2-dehydro-3-deoxy-D-gluconate 5-dehydrogenase (KduD), sorbitol-6-phosphate dehydrogenase (SrlD), NAD(P)-dependent alcohol dehydrogenase (YjgB), ureidoglycolate dehydrogenase (AllD), formate dehydrogenase-*N* (FdnI) (Appendix A), further indicating a probable decreased rate of cell respiration of the ∆*cpxR* mutant. In contrast to the envelope-associated proteins and dehydrogenases, proteins involved in hydrogen metabolism (hydrogenase maturation factor [HybG], hydrogenase-2 assembly chaperone [HybE], hydrogenase expression/formation [HypE], and hydrogenase 3 maturation protein [HypC]) exhibited an increased abundance. Moreover, numerous proteins involved in carbohydrate metabolism showed an increased abundance in the ∆*cpxR* background (Appendix A).

Stress response proteins showed predominantly an increased abundance in the ∆*cpxR* mutant (Appendix A). Several different categories of the stress responses were upregulated in the ∆*cpxR* mutant during the CHX treatment, including oxidative stress response (hydroxylamine reductase [hcP], YkgJ family cysteine cluster protein [YkgJ], thioredoxin domain-containing protein [WP_079957070]; protein repair (ATP-dependent chaperone [ClpB], peptidylprolyl isomerase A [PpiA], protein deglycase [YajL], the bax inhibitor-1/YccA family protein [WP_000373611.1]; and DNA repair (ATP-dependent DNA helicase [DinG], and helix-turn-helix domain-containing protein [TnpR]). Besides the oxidative stress response, DNA and protein repair systems, the ∆*cpxR* mutant upregulated type II toxin-antitoxin systems [CcdB, RelB, and StbD]. A small group of stress response proteins, associated with cold stress [CspC], starvation [PsiF], acid stress [YodD] and ion uptake [MscL], showed a decreased abundance in the ∆*cpxR* mutant (Appendix A).

The CHX treatment caused significant alterations of numerous transporters and receptors in the ∆*cpxR* mutant compared with the wild-type (Appendix A). A great majority (80%) of altered transporters/receptors showed upregulation. Upregulated transporters were exclusively associated with the transenvelope trafficking (uptake) of various types of amino acids and antioxidants, including cysteine/glutathione (CydC), glutamate/aspartate (GltL), D-serine/D-alanine/glycine (CycA), lysine/arginine/ornithine (ArgT), amino acid transporter (GlnH2), and threonine/serine (YjjP). The increased abundance also showed transporters specific for compounds generated by the breakdown of amino acids, putrescine (PlaP), and spermidine (PotF) as well as TonB-dependent receptors (BtuB and WP_001034952.1). In contrast, a small group of sugar transporters (SrlE, WP_001683480.1, and WP_023139385.1) showed a decreased abundance in the ∆*cpxR* mutant. Interestingly, all ribosomal-associated proteins increased in their abundance, whereas all virulence-associated proteins, mainly encoded on *Salmonella* Pathogenicity Island 1, showed a decreased abundance in the ∆*cpxR* mutant during the CHX treatment (Appendix A). It is worth mentioning that numerous proteins/enzymes associated with DNA replication (DnaC), conjugative DNA transfer (TraM), chromosome condensation, segregation (MukB), and DNA methylation exhibited upregulation, indicating the stressful conditions for genome maintenance and replication in the ∆*cpxR* mutant during the CHX treatment.

### 2.7. Validation of Proteomic Data

The expression of ten genes, where protein products showed significant alternations upon the *cpxR* deletion or CHX treatment, were evaluated by qRT-PCR. The selected genes included: *acrB*, *cutC*, *phsA*, *tdcA*, *hlyD*, *rfaC*, *lbpB*, *dedD*, *fumD*, and *pspB*. The qRT-PCR analysis confirmed the proteomic results, showing the same patterns of gene alterations compared with those of their protein products (Figure 6).

## 3. Discussion

We have shown that the *cpxR* deletion significantly increases the susceptibility of *S*. Enteritidis to the antimicrobial agent CHX in the exponential growth phase, whereas deletion of another major envelope stress response regulator, RpoE, causes no change in the pathogen’s susceptibility to this agent. In our earlier study using enterohemorrhagic *E. coli* [23], we revealed that the *rpoE* mutation caused growth phase dependent changes in tolerance to CHX. The *rpoE* mutant, inherently growth deficient, showed reduced susceptibility in the exponential growth phase, whereas during the stationary growth phase (no growth), the same organism exhibited significantly increased susceptibility compared with the wild-type. It was found that the integrity of the lipid bilayer became compromised during the CHX treatment of cells in the stationary growth phase, which led to accumulation of CHX in the *rpoE* mutant and the creation of “leaky-cell” phenotype [23]. The current study confirmed a limiting role of the RpoE regulator in the susceptibility of growing *S*. Enteritidis cells to CXH and showed that in contrast to the RpoE regulator, the CpxR plays a significant role in providing resistance from the CHX treatment to growing *S.* Enteritidis cells. To reveal the overall networks of the CpxR responsible for providing resistance of growing *S.* Enteritidis cells to a polycationic antimicrobial agent, we employed comparative proteomic analyses.

Our study has demonstrated that the deletion of the *cpxR* gene causes extensive physiological changes under a normal, non-stressful growing condition, affecting not only the biogenesis of the envelope and cellular trafficking but also a wide range of critically important biomolecules, including DNA, RNA, and proteins. Raivio et al. [26], inducing the Cpx response with the overproduction of NlpE, an outer membrane protein that activates the extracytoplasmic stress response [26,27], in a combination with comparative transcriptomic analyses, found significant downregulation of genes encoding for oxidative phosphorylation, the Krebs cycle, electron transport, and inner membrane transporters. The same authors found that a large group of genes of unknown function and genes encoding for envelope chaperones and proteases, including *cpxP*, *spy*, *yccA*, and *dsbA*, showed upregulation by NlpE. Although we did not induce the Cpx response by overproduction of NlpE, our data showed that a lack of functional CpxR regulator leads partially to the same physiological changes, including downregulation of the TCA cycle and electron transport as well as upregulation of essential envelope chaperones, including DsbA. Our data also showed that a dysfunctional CpxR regulator causes an increased abundance of proteins associated with biosynthesis of peptidoglycan, lipid A, core LPS, and *O*-antigen as well as proteins associated with recycling of peptidoglycan. Regarding biogenesis and maintenance of the envelope, only proteins associated with peptidoglycan maturation and remodelling exhibited decreased abundance in the ∆*cpxR* background. The *cpxR* deletion changed cell trafficking, generally increasing the abundance of transporters critical for membrane integrity, uptake of amino acids and large molecules, whereas decreasing abundances of transporters that were associated with the TCA cycle and electron transport. Moreover, major porin, OmpC, and two additional porins, OmpN and Tsx, showed decreased abundance in the *cpxR* background, clearly indicating that deletion of the *cpxR* gene imposes a stressful condition for the mutant cell. It is well documented that the decrease of porin abundance in a prokaryotic cell is an adaptive response of these organisms to exposure to a wide spectrum of antimicrobials [23,28,29,30,31] and reactive oxygen species (ROS) [32]. Guest et al. [11] demonstrated that the Cpx response affects the respiratory complexes NADH dehydrogenase I and cytochrome *bo*_3_. Furthermore, the same group [11] hypothesized that damage to the inner membrane, where these respiratory complexes are located, could generate harmful ROS which can lead to irreversible aggregation of proteins. Our data indicated that the ∆*cpxR* mutant is probably exposed to endogenous ROS. Not only a wide spectrum of DNA repair proteins, both envelope, and cytosol chaperones and proteases were induced, but also powerful antioxidants, including thioredoxin, glutaredoxin 1, glutaredoxin 3, and disulphide reductase, hallmarks of the oxidative stress response [32,33,34,35], showed increased abundance in the *cpxR* background. Taken together, decreased abundance of porins and increased abundance of antioxidants, DNA repair systems, envelope, and cytosol chaperones, and proteases, provide multiple pieces of evidence that the ∆*cpxR* mutant is under stress and probably exposed to an increased level of endogenous ROS.

As both the CpxR regulator and antibacterial agent CHX are functionally and therapeutically associated with the cellular membrane, we first examined their effects on membrane integrity and cellular turgor. Using a series of TEM analyses, it has been shown that neither the CHX treatment nor *cpxR* deletion nor a combination of CHX treatment and *cpxR* deletion could cause any visible changes to the membrane integrity and cellular turgor. Tattawasart et al. [36], using a similar approach, found that the CHX treatment caused disorganization of the outer membrane, a significant loss of cytoplasmic material and finally lysis of Gram-negative organism, *Pseudomonas stutzeri*. It is worth noting that Tattawasart et al. [36] used 100 mg/L concentration of CHX, whereas in our study we used 3 mg/L, as this concentration proved to be a MIC for our model organism. It is important to emphasize that the objectives of these two studies were different. While Tattawasart et al. [36] aimed to reveal the effect of CHX on the cellular structure, we aimed to reveal the role of the CpxR regulator in the susceptibility of *Salmonella* to CHX. Importantly, our TEM analyses showed that the initial lethality of the *cpxR* mutant treated with MIC of CHX was not associated with the loss of cellular turgor or major membrane perturbation but rather with less invasive mechanisms.

During the CHX treatment, the proteome of the ∆*cpxR* mutant underwent some similar changes to those during the non-treatment regime. In both cases, during CHX treatment and non-treatment, the ∆*cpxR* mutant altered cellular trafficking in the same way, increasing the abundance of amino acids and TonB-dependent receptors and decreasing the abundance of sugar (TCA cycle), electron transporter receptors and porins. Similarly, a wide spectrum of DNA repair proteins, chaperones, proteases, and antioxidants showed increased abundance in the ∆*cpxR* background during the CHX treatment. Our study showed that alteration of cellular trafficking and stress responses, although increased during CHX treatment, are in general intrinsically associated with a dysfunctional CpxR regulator. Besides these common physiological changes, the ∆*cpxR* mutant induced specifically a group of type II toxin-antitoxin systems only during the CHX treatment. Type II toxin-antitoxin systems are relatively small proteins made of a protein that causes toxic effect and its counteracting protein that neutralizes this toxicity [37]. It has been shown that these systems are involved in a multifaceted stress response, including genomic stabilization [38], biofilm formation [39], oxidative stress response [37], starvation response [40], and generation of persister cells able to survive antimicrobial treatment [41]. This study showed that *S*. Enteritidis with no functional CpxR regulator during the CHX treatment induces several type II toxin-antitoxin systems to modulate responses to both the antiseptic treatment and the lack of functional CpxR regulator, further indicating the importance of these toxin-antitoxin systems in the response of *S*. Enteritidis to antiseptic treatment. Moreover, specific proteome changes during the CHX treatment were associated with the envelope itself. The ∆*cpxR* mutant exhibited decreased abundance of enzymes and proteins associated with the recycling of muropeptides and resistance to polymixin and antimicrobial peptides. In addition, it is worthy of mentioning that certain proteins, including RelB, ClpB, and ribonuclease E (rne), showed an outstanding increase in abundance compared to other altered proteins in their groups, DNA repair, chaperone and folding catalysts and RNA degradation, respectively. It is important to emphasize that one of the hallmarks of the stress response chaperones, ClpB, showed an outstanding increase in abundance, indicating an increased level of aggregated proteins in the ∆*cpxR* background.

In conclusion, to the best of our knowledge, this is the first study that reports the overall effect of a dysfunctional CpxR regulator on the physiology of growing prokaryotic cells. We have shown that a lack of this extracytoplasmic stress response regulator induces multifaceted stress responses not only in the envelope but also in the cytosol, further affecting the key biomolecules including DNA, RNA, and proteins. Our group was also able to separate the effect of the *cpxR* deletion from that of the CHX treatment, showing that alterations in the cellular trafficking and most of the stress responses are intrinsically associated with the lack of the CpxR regulator. Induction of type II toxin-antitoxin systems, and decrease in abundance of enzymes and proteins associated with the recycling of muropeptides and resistance to polymixin and antimicrobial peptides, were specific responses of the ∆*cpxR* mutant to the CHX treatment. Collectively, our study provides novel insight into the effects of CpxR on the physiology of *S*. Enteritidis cells during normal, non-stressful growing conditions and with the CHX treatment, which may point to potential cellular targets for the development of an effective antimicrobial agent.

## 4. Material and Methods

### 4.1. Bacterial Strains, Plasmids and Growth Conditions

A host for the recombinant plasmid pTre99A::*cpxR* was *Escherichia coli* K-12 strain DH5, while *Salmonella enterica* serovar *Enteritidis* strain ATCC 13076 was used as the parental organism. Plasmid pTre99A served as an expression vector and plasmid pKD3 was used as a source to amplify the chloramphenicol cassette. Plasmid pKD46 and temperature-sensitive plasmid pCP20 were regularly used during the recombineering procedure. Antibiotics, ampicillin (100 µg/mL), chloramphenicol (30 µg/mL), sugar, arabinose 10 mM (Sigma Chemical Co., St Louis, MO, USA) were added to Luria-Bertani (LB) medium to maintain or to select mutant strains.

### 4.2. Construction of Isogenic ΔrpoE and ΔcpxR Mutants

In frame deletions of the *rpoE* and *cpxR* genes were carried out using the Red Lambda recombineering approach as described previously [42,43]. In short, the pKD3 plasmid was used as a source for a chloramphenicol (*cat*)-resistant cassette. The wild-type strain containing pKD46 plasmid was transformed with amplified *cat* cassettes with 50-nucleotide (nt) tails that are homologues to the region of 50 nt upstream and downstream of the target gene. Deletion of the targeted gene was confirmed by PCR and DNA sequencing. The *cat* cassette from the Δ*rpoE* and Δ*cpxR* mutants was excised by the introduction of a temperature-sensitive pCP20 plasmid. The final *rpoE* and *cpxR* knockouts were selected at 42 °C and their deletions were verified by PCR and DNA sequencing.

### 4.3. Cell Growth Assay

Cultures of frozen wild-type, ∆*rpoE,* and ∆*cpxR* strains were used to streak out Luria-Bertani (LB) agar plates. Inoculated plates were incubated overnight at 37 °C. Three single colonies of each strain were used to inoculate 50 mL of LB. The inoculated media was incubated for 24 h at room temperature (22 °C) with permanent shaking at 190 rpm. After this, cultures were diluted 100 fold followed by incubation at room temperature with permanent shaking at 190 rpm. Cell growth was measured based on the increase of biomass to optical densities at 600 nm over the ten hours of the incubation period.

### 4.4. Determination of the Minimum Inhibitory Concentration of Chlorhexidine

The MIC of CHX against the wild-type of *S.* Enteritidis was determined first by using a wide range of antiseptic concentrations, including 25, 20, 15, 10, 5, and 1 μg/mL. Once the effective ranges were identified, a narrower concentration range (4, 3, and 2 μg/mL) was used to pinpoint a concentration of CHX that exhibits an inhibitory effect on the growing culture of *S.* Enteritidis. Determination of the MIC was completed over the 180-min CHX exposure time course as described below. The only difference was that in this assay instead of viable cell counts, bacterial growth and survivability during the CHX treatment was determined by optical density measurements at 600 nm. Once the MIC of CHX was determined, it was used in the subsequent experiments.

### 4.5. Drug Treatment Assay

A CHX treatment was performed using freshly overnight-cultured wild-type strain and its two isogenic ∆*rpoE* and ∆*cpxR* mutants on LB agar plates at 37 °C. Several single colonies of each organism were inoculated into 50 mL of LB, followed by overnight incubation at 37 °C with constant shaking at 190 rpm. The seed cultures were diluted 1/100 in 100 mL of LB and incubated at 37 °C to optical densities at 600 nm of 0.4. Once these cultures reached their midpoint of the exponential growth phase (0.4), samples were challenged with CHX at a final concentration of 3 µg/mL. The CHX challenged cultures were incubated for 3 h at 37 °C with permanent shaking at 190 rpm. Immediately before CHX challenge and then every 30 min of the CHX treatment, cell counts were carried out using tenfold serial dilutions. Diluted samples of 0.1 mL were plated on LB agar in triplicate followed by incubation at 37 °C overnight.

### 4.6. Complementation Study

The *cpxR* gene sequence was amplified by PCR using the DNA templets of the wild-type strain, Q5 DNA polymerase (New England BioLabs, Ipswich, MA, USA). The following primers Ps (5′-*TGG* GAA TTC CAT ATG AAT AAA ATC CTG TTA GTT G-3′), Pas (5′-*CCC* AAG CTT TCA CTT GTC ATC GTC TTT GTA GTC TGA AGC GGA AAC CAT CAG ATA G-3′) were used. The primer clamps are shown in italics and the EcoRI, HindIII sites, and red flag are underlined. The amplified *cpxR* gene and pFPV plasmid were each double digested using EcoRI and HindIII restriction enzymes. The digested *cpxR* gene sequence was cloned into pFPV plasmid. The Δ*cpxR* mutant was transformed with the recombinant pFPV::*cpxR* plasmid and Δ*cpxR*-complemented mutant strain was selected on LB agar containing ampicillin. Furthermore, the wild-type strain was transformed with an empty pFPV vector. All four strains, including the wild-type, Δ*cpxR* mutant, Δ*cpxR* mutant complemented with a functional *cpxR* gene and the wild-type strain transformed with the empty pFPV vector, were used to find out the effect of *cpxR* deletion on the ability of *S. enteritidis* to withstand exposure to CHX.

### 4.7. Morphological Analysis

The samples of the wild-type and Δ*cpxR* mutant strains were prepared and treated with CHX as described above. Samples of the wild-type and Δ*cpxR* mutant were taken after 60 min of CHX treatment and with no CHX exposure, following centrifugation at 2000× *g.* After this samples were fixed using 2% glutaraldehyde and 2.5% paraformaldehyde in 0.1 M sodium cacodylate (NaCac) buffer for at least 2 h at room temperature, followed by incubation overnight at 4 °C. Samples were rinsed in 0.1M NaCac buffer for 10 min three times, then put in secondary fix (1% osmium tetroxide in 0.1M NaCac buffer) overnight at 4 °C. After this, samples were rinsed in ultrapure water (10 min, 3 times), then buried in low temperature melting agarose and cut into approximately 1 mm^3^ pieces. Material was dehydrated in an ethanol using a series of concentrations, including (25%, 50%, 75%, 95% [10 min, 2 times], 100% [10 min, 3 times]), infiltrated with Embed-812 resin (1:1 resin: ethanol, 100% with hardener [twice]), and placed in a 60 °C oven for 48 h [44]. Material was cut using a diamond knife on a Leica Ultracut UCT microtome at a thickness of 70–100 microns then collected on 200-mesh formvar/carbon-coated copper grids followed by staining with 3% aqueous uranyl acetate (20 min) and Sato’s lead citrate stain (3 min). Grids were observed on a JEOL JEM-1400 Plus transmission electron microscope operating at 120 kV. Images were captured with an Advanced Microscopy XR16 camera.

### 4.8. Samples Preparation for Protein Analysis

The CHX-treated and CHX-untreated 10 mL cultures of the wild-type and ∆*cpxR* mutant were sampled after 30 min of incubation and/or CHX treatment followed centrifugation at 13,000× *g* for 5 min. The resulting culture pellets were washed two times and frozen at −80 °C.

### 4.9. Protein Extraction, Proteolytic Digestion and iTRAQ Labelling

The cellular proteins were extracted from resuspended cell pellets in 400 µL of extraction buffer [7 M urea, 2 M thiourea, 0.4 M triethylammonium bicarbonate (TEAB), pH 8.5, 20% acetonitrile, and 4 mM Tris (2-carboxyethyl) phosphine hydrochloride (TCEP)]. The extraction solutions were incubated at 37 °C for 1 h, followed by the addition of methyl methanethiosulfonate to a concentration of 8 mM. The cell extraction solutions with suspended bacterial materials were centrifuged at 10,000× *g* for 5 min. The supernatants were transferred to sterile microcentrifuge tubes followed by determination of protein concentration with a Bradford assay and standardization of the samples’ protein concentrations. Protein digestion was carried out as previously described [45,46]. The iTRAQ labelling of tryptic peptides derived from the wild-type and mutant strains with CHX treatment and no-CHX treatment conditions was carried out using iTRAQ reagent multiplex kit (Applied Biosystems, Foster City, CA, USA) according to manufacturers’ recommendations. The labelled peptide samples were pooled and dried prior to high-pressure liquid chromatography (HPLC) fractionation. The HPLC fractionation was carried out as previously described [47]. Three biological replications were carried out for both the wild-type and *cpxR* mutant treated with CHX, while two biological replications were performed for the wild-type and its *cpxR* mutant with no CHX treatment. Each sample was labelled with a different isobaric tag and then processed according to the previously described method [47].

### 4.10. Peptide LCMS Analysis

We analyzed approximately 400 ng of reconstituted peptide aliquots from the first dimension LC fractions by capillary LC-MS on an Orbitrap Velos MS system as previously described [47] with some modifications. Briefly, the capillary column diameter was 100 µm, the gradient elution profile was of 8–35% B Solvent over 67 min at 330 nanoliters/min. A Solvent was 98:2:0.01, H_2_O: acetonitrile (ACN): formic acid (FA); and B Solvent was 98:2:0.01, ACN: H2O: FA, HCD (higher-energy collision-induced dissociation). Activation time was 20 ms and lock mass was not employed. Dynamic exclusion settings were the following, repeat count = 1, exclusion list size was 200, exclusion duration = 30 s, and exclusion mass width (high and low) was 15 ppm. Early expiration was disabled. FT MS1 injection time was 300 ms.

### 4.11. Database Searching with Protein Pilot

The *.RAW to *.MGF conversion was described in Lin-Moshier et al. [47]. Protein Pilot 5.0.1.0 (Sciex) searches with the Paragon Search Algorithm v 5.0.1.0 were performed against the NCBI Reference Sequence *Salmonella enterica* subsp. *enterica* serovar *enteritidis* strain P125109 (accession AM933172; downloaded 18 September 2017) protein sequence database (49,348 proteins), to which a contaminant database (thegpm.org/crap/index, 109 proteins) was appended. Search parameters were: cysteine alkylation MMTS; digestion enzyme trypsin; instrument Orbi MS (1–3 ppm) Orbi MS/MS; biological modifications ID focus; thorough search effort; and False Discovery Rate analysis (with reversed database), bias correction applied.

### 4.12. Quantitative Data Analysis

Normalization was carried out across samples and spectra as previously described [48]. The medians were used for averaging. Spectra data were log-transformed, pruned of those matched to multiple proteins and those missing a reference value and weighted by an adaptive intensity weighting algorithm. Different abundances of proteins in various samples compared with the average of the pooled control samples were determined by Scaffold Q + by applying a permutation test with the significance threshold set at a *p* value of <0.05.

### 4.13. Validation of Proteomic Data by qRT-PCR

The proteomic data were verified by quantitative real-time PCR (qRT-PCR). Samples for the qRT-PCR analysis were taken same as for protein analysis. The RNeasy Mini kit (Qiagen Inc., Valencia, CA, USA) was used to extract RNAs according to the manufacturer’s instructions. The iScript™ Reverse Transcription (Bio-Rad Laboratories, Inc. Hercules, CA, USA) was used to synthetize cDNA. qRT-PCR was carried out as described previously [49], on a MiniOpticon™ Real-Time PCR Detection System (Bio-Rad Laboratories, Hercules, CA, USA). The iQTM SYBR Green Supermix kit (Bio-Rad Laboratories, Hercules, CA, USA) was used. For internal reference, *rtcR*, was selected. During the qRT-PCR assay optimization, it was observed that the *rtcR* transcripts in the wild-type and *cpxR* mutant were not affected by antimicrobial agent or a gene deletion. Primers, designed to measure expression of ten genes, including *acrB*, *cutC*, *phsA*, *tdcA*, *hlyD*, *rfaC*, *ibpB*, *dedD*, *fumD* and *pspB*, are presented in Table 3.

### 4.14. Experimental Replication and Bioinformatics

Data from the cell growth, determination of MIC of CHX, antiseptic treatment assays, complementation study and qRT-PCR are the average of three replications. The iTRAQ data related to CHR treatments of the wild-type and *cpxR* mutant represent also the means of three replicates, while iTRAQ data relevant to the no-CHX treatment represent the averages of two replications. The kinetic data for growth rates and antiseptic treatment assay were analyzed with a two-way analysis of variance (ANOVA), with strain and times as factors, followed by Tukey’s *post hoc* test at the 5% level. The residuals were checked to see that the assumptions behind the ANOVA model were satisfied. The analyses were carried out using Genstat 18th
edition (VSN International, Hemel Hempstead, UK). A value of *p* < 0.05 was considered statistically significant. Gene expression data were analyzed by *t*-test. The *p* values for the differences in protein abundance of different samples (wild-type vs *cpxR* mutant and CHX treatment vs no CHX treatment) were determined using a two-tailed Student’s *t*-test. The Database for Annotation, Visualization, and Integrated Discovery (DAVID) [50] and KEGG Mapper–Search Pathway were used to perform gene ontology analysis.

## Figures and Tables

**Figure 1 ijms-22-08938-f001:**
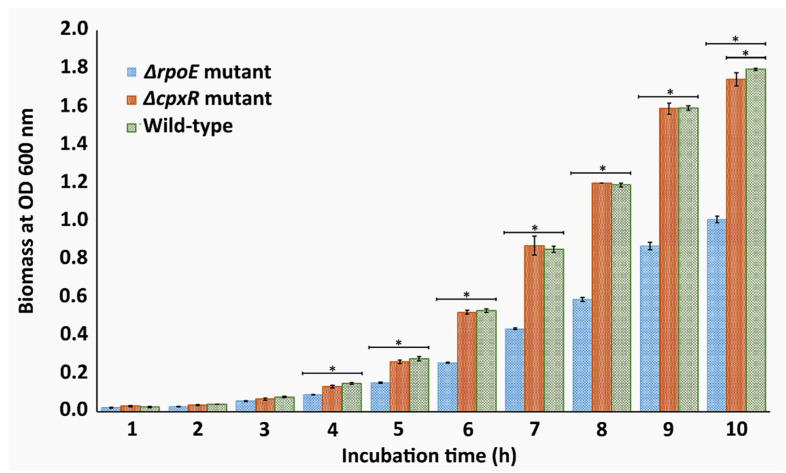
Evaluation of the effects of *cpxR* and *rpoE* deletions on the growth kinetics of *S. enteritidis* during the exponential and early stationary growth phases. Seed cultures were diluted 100 folds and incubated at 22 °C with permanent shaking at 190 rpm. The growth of cultures was measured using optical density at 600 nm for every 60-min during the 10-h period. The error bars indicate the standard errors of the means (*n* = 3). Asterisks (*) represent statistically significant differences (*p* < 0.05). Data are averages of three biological replications.

**Figure 2 ijms-22-08938-f002:**
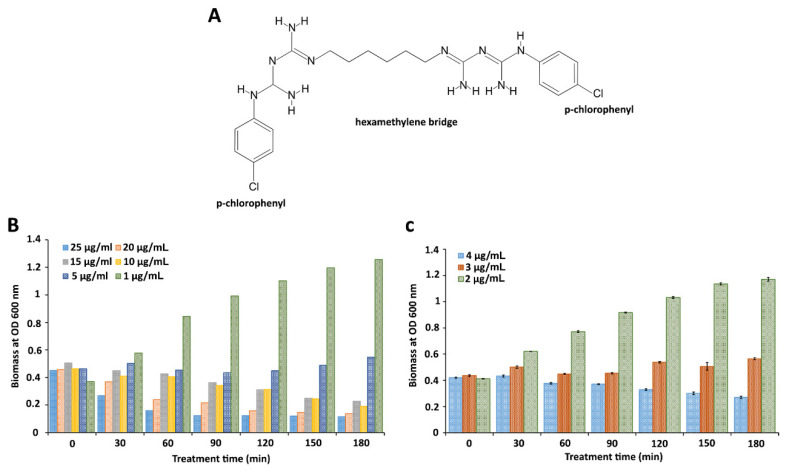
Determination of the MIC of chlorhexidine for the wild-type *S.* Enteritidis. (**A**) Molecule of chlorhexidine, composed of two p-chlorophenyl rings connected by central hexamethylene bridge. (**B**) Freshly grown *S.* Enteritidis cultures with an optical density at 600 nm of 0.4 (~1 × 10^8^ cell forming units per mL) were treated with a wide range of chlorohexidine concentrations. Viable cell counts are shown for every 30-min treatment during 180 min. Data represent one biological replication. (**C**) The same treatment approach, with a narrow range of chlorhexidine concentrations (2, 3, and 4 µg/mL), was used in combination with three biological replications. The error bars indicate the standard errors of the means.

**Figure 3 ijms-22-08938-f003:**
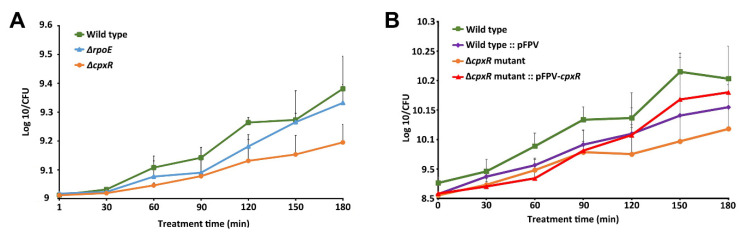
Chlorhexidine killing assay. (**A**) The effect of MIC (3 µg/mL of CHX) on ∆*rpoE* and ∆*cpxR* mutant *S.* Enteritidis strains compared with the wild-type over the exponential growth phase. (**B**) The effect of the *cpxR* deletion on the susceptibility of *S.* Enteritidis to chlorhexidine as well as the susceptibility of respective complemented strains.

**Figure 4 ijms-22-08938-f004:**
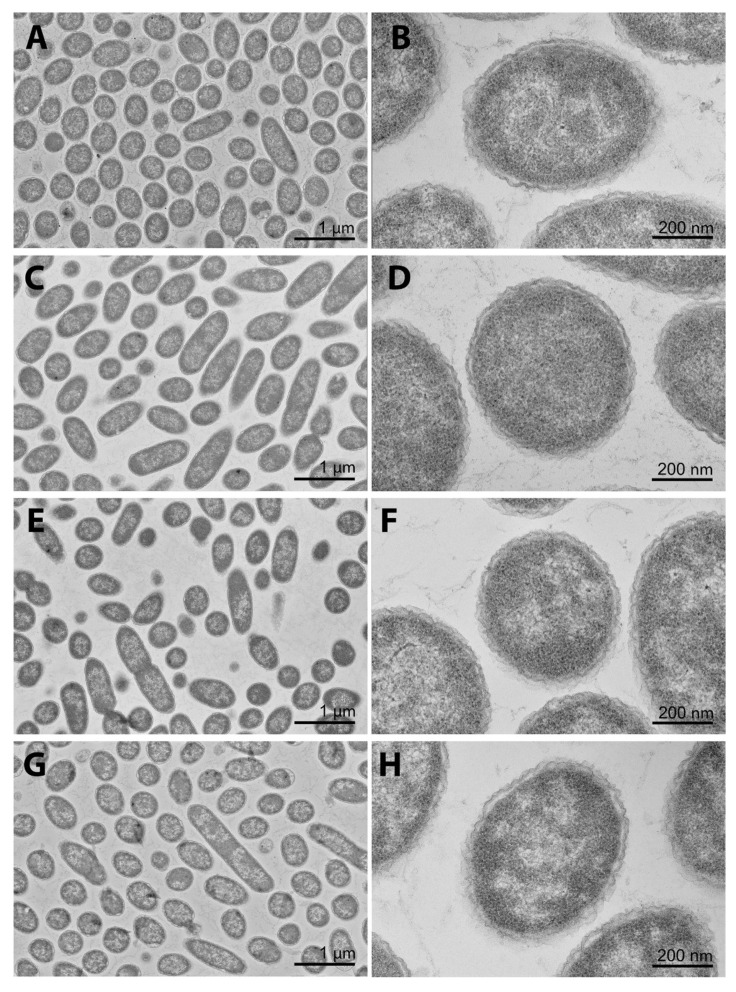
Representative TEM images, taken after 60 min with and without chlorhexidine treatment of the wild-type and ∆*cpxR* mutant of *S*. Enteritidis, respectively. Images (**A,B**) portray wild-type cells with no chlorhexidine treatment, while images (**C**,**D**) show wild-type cells during the chlorhexidine treatment. Similarly, images (**E**,**F**) depict the ∆*cpxR* mutant during no treatment, whereas images (**G**,**H**) show the same organism during the chlorhexidine treatment.

**Figure 5 ijms-22-08938-f005:**
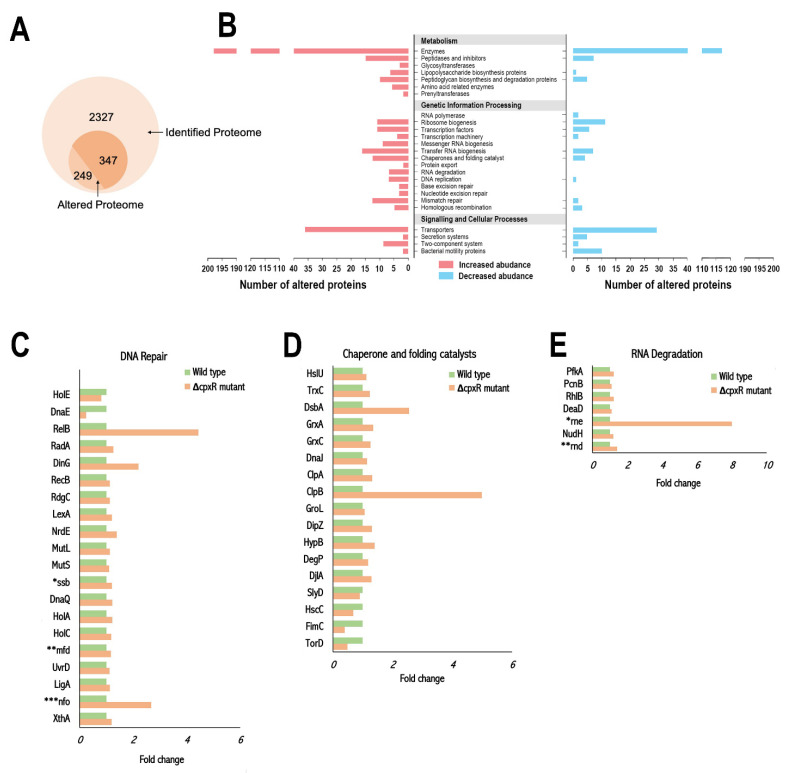
Effect of the *cpxR* deletion on the proteome of *S*. Enteritidis during the exponential growth phase. (**A**) Venn diagram showing the overall effect of the *cpxR* deletion on the proteome of *S.* Enteritidis. Approximately, out of 2327 identified proteins, one quarter (25.6%) of the proteome was altered. There were 347 proteins with increased abundance and 249 proteins with decreased abundance compared with that of the wild-type. (**B**) Protein ontology enrichment analysis portraying the biological processes that have been altered by the deletion of the *cpxR* gene. All proteins that showed statistically significant differences, including the permutation test and Benjamini-Hochberg method (*p* < 0.023), as well as high reproducibility across the biological replications were analyzed, (**C**–**E**). Fold change in protein abundances of proteins involved in (**C**) DNA repair, * single-stranded DNA-binding protein SSB1, ** transcription-repair coupling factor, *** deoxyribonuclease IV (**D**) chaperone and folding catalysts and (**E**) RNA degradation, * ribonuclease (**E**), ** ribonuclease (**D**). These three graphs indicate the alterations of these three groups of proteins in the ∆*cpxR* background compared with the wild-type during the exponential growth phase.

**Figure 6 ijms-22-08938-f006:**
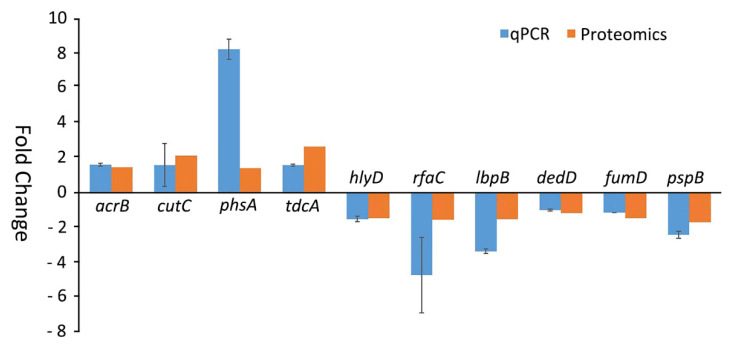
Validation of proteomic (HPLC-iTRAQ) data by qRT-PCR analysis. The proteomic data of selected proteins (AcrB, CutC, PhsA, TdcA, HlyD, RfaC, IbpB, DedD, FumD and PspB) were evaluated by comparing their abundances with the expression levels of their gene under the same conditions.

**Table 1 ijms-22-08938-t001:** Differential abundance of proteins in the ∆*cpxR* background compared with that of the wild-type *S*. Enteritidis during the exponential growth phase.

KO Identifiers	Gene Name	Accession Number	Mol. wt. kDa	Benjamini Hochberg *p* Value	Fold Change	Description
**Peptidoglycan Biosynthesis** **and Degradation**						
K00075	*murB*	WP_000149793.1	38	0.00012	1.27	UDP-*N*-acetylmuramate dehydrogenase
K01928	*murE*	WP_000775071.1	53	0.001	1.15	UDP-*N*-acetylmuramoyl-l-alanyl-d-glutamate--2,6-diaminopimelate ligase
K01776	*murL*	WP_031619622.1	31	0.013	1.12	Glutamate racemase
K01297	*ldcA*	WP_000051603.1	33	0.00078	1.23	Muramoyltetrapeptide carboxypeptidase
K01921	*ddl*	WP_000763905.1	33	0.001	1.26	d-alanine-d-alanine ligase
K03806	*ampD*	WP_000936324.1	21	0.003	1.20	*N*-acetyl-anhydromuramoyl-l-alanine amidase
K06078	*lpp*	WP_001082307.1	8	0.021	1.18	Murein lipoprotein
K08306	*mltC*	WP_000976287.1	40	0.006	1.32	Membrane-bound lytic murein transglycosylase C
K19236	*ycfS*	WP_001708681.1	33	0.002	2.22	l,d-transpeptidase YcfS
K01448	*amiA*	WP_069040808.1	32	0.002	0.81	*N*-acetylmuramoyl-l-alanine amidase
K01775	*alr*	WP_001147296.1	39	0.0016	0.90	Alanine racemase
K07258	*dacD*	WP_000925044.1	43	0.002	0.40	Serine-type d-Ala-d-Ala carboxypeptidase
K18988	*ampH*	WP_000830784.1	42	0.004	0.46	Serine-type d-Ala-d-Ala endopeptidase
K21470	*ltdT*	WP_000925899.1	68	0.002	0.72	l,d-transpeptidase
**Lipopolysaccharide Biosynthesis**						
K00677	*lpxA*	WP_000565950.1	28	0.003	1.32	UDP-*N*-acetylglucosamine acyltransferase
K02517	*lpxL*	WP_000163977.1	35	0.01	1.21	Kdo2-lipid IVA lauroyltransferase/acyltransferase
K02535	*lpxC*	WP_000595474.1	34	0.012	1.23	UDP-3-*O*-[3-hydroxymyristoyl] *N*-acetylglucosamine deacetylase
K12979	*lpxO*	WP_000457031.1	35	0.012	1.50	Lipid A hydroxylase LpxO
K02841	*rfaC*	WP_076915342.1	35	0.00031	1.57	Lipopolysaccharide heptosyltransferase
K03760	*eptA*	WP_038427769.1	62	0.013	1.25	Lipid A ethanolaminephosphotransferase
K10011	*arnA*	WP_038425381.1	73	0.02	1.10	UDP-4-amino-4-deoxy-l-arabinose formyltransferase/UDP-glucuronic acid dehydrogenase
K09953	*lpxR*	WP_001046434.1	35	0.002	0.44	Lipid A 3-*O*-deacylase
**Glycosyltransferases**						
K02841	*cobK*	WP_076915342.1	35	0.00032	1.27	Lipopolysaccharide heptosyltransferase
K02852	*rffM*	WP_000183613.1	28	0.012	1.32	Lipopolysaccharide *N*-acetylmannosaminouronosyltransferase
K12992	*rfbN*	WP_000705151.1	36	0.00012	1.16	O antigen biosynthesis rhamnosyltransferase
**Transporters**						
K02000	*proV*	WP_069057539.1	44	0.017	1.23	Glycine betaine/proline transporter
K02068	*fetA*	WP_000166987.1	25	0.004	1.21	Iron ABC transporter ATP-binding protein
K02071	*metN*	WP_079837681.1	32	0.00018	1.30	Methionine ABC transporter
K02598	*nirC*	WP_000493575.1	29	0.004	1.69	Nitrite transporter
K02759	*chbA*	WP_001732541.1	11	0.001	1.23	Cellobiose PTS system EIIA component
K03475	*ulaA*	WP_001721663.1	47	0.004	1.50	Ascorbate PTS system EIIC component
K03561	*exbB*	WP_000527859.1	26	0.004	1.28	Biopolymer transporter ExbB
K03562	*tolQ*	WP_000131318.1	26	0.001	1.33	Biopolymer transport protein TolQ
K03832	*tonB*	WP_001517937.1	26	0.006	1.19	TonB system transport protein TonB
K04759	*feoB*	WP_000736978.1	84	0.015	1.15	Ferrous iron transport protein B
K05685	*macB*	WP_000125899.1	71	0.003	1.22	Macrolide ABC transporter permease MacB
K05776	*modF*	WP_079983686.1	55	0.001	1.27	Molybdate transporter ATP-binding protein
K06155	*gntT*	WP_001131737.1	46	0.012	1.77	Gluconate transporter
K07091	*lptF*	WP_000584130.1	40	0.005	1.29	LPS export ABC transporter permease LptF
K07127	*hiuH*	WP_080199438.1	15	0.004	1.32	5-hydroxyisourate hydrolase
K07278	*tamA*	WP_001120233.1	65	0.00016	1.26	Translocation and assembly module TamA
K07308	*dmsB*	WP_132631290.1	55	0.00045	1.34	Dimethylsulfoxide reductase subunit B
K08351	*bisC*	WP_023227718.1	86	0.006	1.17	Biotin/methionine sulfoxide reductase
K09013	*sufC*	WP_001580259.1	28	0.0004	1.20	Fe-S cluster assembly ATP-binding protein
K09810	*lolD*	WP_001033714.1	25	0.004	1.35	Lipoprotein-releasing ABC transporter
K09812	*ftsE*	WP_000617729.1	24	0.002	1.26	Cell division transporter ATP-binding protein
K09817	*znuC*	WP_000203014.1	28	0.001	1.22	Zinc ABC transporter ATP-binding protein
K09997	*artJ*	WP_000756583.1	27	0.002	1.11	Arginine ABC transporter
K10001	*gltI*	WP_000588819.1	34	0.003	1.30	Glutamate/aspartate transporter
K10017	*hisP*	WP_000986780.1	29	0.012	1.49	Histidine transporter ATP-binding protein
K10439	*rbsB*	WP_023254734.1	31	0.001	1.13	Ribose transporter substrate-binding protein
K10540	*mglB*	WP_069057568.1	36	0.00056	1.27	Methyl-galactoside transporter
K10542	*mglA*	WP_000535907.1	56	0.004	1.64	Methyl-galactoside transporter
K11073	*potF*	WP_000125769.1	41	0.004	1.43	Putrescine transporter
K11720	*lptG*	WP_001182241.1	40	0.009	1.26	LPS export ABC transporter permease
K13893	*yejA*	WP_135416368.1	69	0.0049	1.39	Microcin C substrate-binding protein
K16012	*cydC*	WP_001202251.1	63	0.0055	1.39	Cysteine/glutathione ABC transporter
K18138	*adeB*	WP_001132506.1	114	0.001	1.27	Multidrug efflux RND transporter
K19226	*sapA*	WP_001241619.1	62	0.001	1.27	Peptide ABC transporter substrate-binding
K23991	*ptsP*	WP_079829802.1	85	0.003	3.46	Multiphosphoryl transfer protein
K24163	*nhaK*	WP_001696843.1	60	0.016	1.10	Na+/H+ antiporter
K01999	*livK*	WP_000676957.1	39	0.003	0.48	Branched-chain amino acid ABC transporter
K02017	*modC*	WP_000891715.1	39	0.00037	0.61	Molybdate transporter ATP-binding protein
K02036	*pstB*	WP_000063118.1	29	0.00017	0.82	Phosphate ABC transporter
K02064	*tbpA*	WP_000915326.1	36	0.00083	0.88	Thiamine ABC transporter
K02445	*glpT*	WP_080195040.1	44	0.004	0.25	Glycerol-3-phosphate transporter
K02575	*narK*	WP_000019850.1	50	0.001	0.55	Nitrate transporter NarK
K02774	*gatB*	WP_000723161.1	10	0.004	0.19	PTS sugar transporter subunit IIB
K02777	*crr*	WP_000522253.1	18	0.014	0.87	PTS glucose transporter subunit IIA
K02779	*ptsG*	WP_000475705.1	50	0.001	0.78	PTS glucose transporter subunit IIBC
K02782	*srlE*	WP_000199033.1	34	0.007	0.57	PTS glucitol/sorbitol transporter subunit IIB
K02784	*ptsH*	WP_000487600.1	9	0.002	0.76	Phosphocarrier protein Hpr
K03284	*corA*	WP_000947139.1	37	0.014	0.86	Magnesium/cobalt transporter CorA
K04758	*feoA*	WP_061451033.1	8	0.01	0.73	Ferrous iron transporter A
K05517	*tsx*	WP_000752021.1	33	0.004	0.74	Nucleoside-specific channel-forming protein
K05816	*ugpC*	WP_000907837.1	39	0.004	0.80	Sn-glycerol-3-phosphate import protein UgpC
K05845	*opuC*	WP_000155871.1	33	0.002	0.40	ABC transporter substrate-binding protein
K07122	*mlaB*	WP_000188843.1	11	0.018	0.79	Lipid asymmetry maintenance protein MlaB
K07306	*dmsA*	WP_077917388.1	89	0.004	0.10	Anaerobic dimethyl sulfoxide reductase A
K07795	*tctC*	WP_000744418.1	30	0.002	0.40	Tricarboxylic transport membrane protein
K08154	*emrD*	WP_000828735.1	42	0.007	0.51	Multidrug efflux MFS transporter EmrD
K08353	*phsB*	WP_001015351.1	21	0.003	0.47	Thiosulfate reductase electron transport protein
K09475	*ompC*	WP_000758335.1	41	0.002	0.50	Outer membrane pore protein C
K10555	*lsrB*	WP_079920817.1	37	0.002	0.73	AI-2 transport substrate-binding protein
K11732	*pheP*	WP_000786283.1	51	0.02	0.75	Phenylalanine-specific permease
K11738	*ansP*	WP_000857110.1	54	0.0015	0.58	l-asparagine permease
K14062	*ompN*	WP_000824321.1	42	0.019	0.35	Outer membrane protein N
K18141	*acrE*	WP_000160380.1	41	0.02	0.74	Efflux RND transporter periplasmic adaptor
K23188	*fepC*	WP_023227378.1	29	0.004	0.51	Iron-enterobactin ABC transporter

**Table 2 ijms-22-08938-t002:** Alteration of membrane-associated proteins in the *S*. Enteritidis ∆*cpxR* mutant due to CHX treatment.

Gene Name	Accession Number	Mol. wt. kDa	Benjamini Hochberg *p* Value	Log 2 Fold Change	Fold Change	Description
*emtA*	WP_000776974.1	22	0.006	−0.85	0.55	Membrane-bound lytic murein transglycosylase
*asmA*	WP_023227411.1	75	0.006	−1.43	0.37	Assembly of outer membrane proteins
*frlB*	WP_023206877.1	37	0.006	−3.08	0.12	Fructoselysine-6-*P*-deglycase
*arnC*	WP_000458893.1	37	0.001	−0.84	1.55	Undecaprenyl-phosphate 4-deoxy-4-formamido-l-arabinose transferase
*pepE*	WP_000421776.1	25	0.018	−1.55	0.34	Dipeptidase
*fdxH*	WP_000061599.1	32	0.001	−1.13	0.46	Formate dehydrogenase subunit beta
*hemX*	WP_000138954.1	42	0.001	−1.83	0.28	Uroporphyrinogen-III *C*-methyltransferase
*nfsB*	WP_000355870.1	24	0.006	−4.38	0.05	Oxygen-insensitive NAD(P)H nitroreductase
	WP_065618791.1	27	0.006	−1.01	0.50	Phosphatase PAP2 family protein
	WP_000750393.1	8	0.006	−1.13	0.46	YgdI/YgdR family lipoprotein
	WP_000748128.1	11	0.006	−3.64	0.08	EexN family lipoprotein
	WP_001095011.1	58	0.006	−2.36	0.19	Membrane protein
	WP_001240360.1	39	0.00017	−1.6	0.33	Membrane protein
*rfbB*	WP_000697848.1	41	0.006	3.87	14.62	dTDP-glucose 4,6-dehydratase
*malS*	WP_000761323.1	76	0.006	1.17	2.25	Periplasmic alpha-amylase

**Table 3 ijms-22-08938-t003:** Primers used for evaluation of proteomic iTRAQ data.

Gene Name.	Protein Function	Primer Pair	Sequence of Primers
*acrB*	Multidrug efflux RND transporter AcrB	*acrB*–F	CACGAAACCAATCTGCGTAAAG
		*acrB*–R	CTTCGCCGTCCTGCTTATT
*cutC*	Copper homeostasis protein CutC	*cutC*–F	GAATACTCGTCCGCCTGTATATC
		*cutC*–R	GAGTACGGGAAGTACACAGTTC
*phsA*	Thiosulfate reductase PhsA	*phsA*–F	ATCTCATCGCCGGTCTTAATG
		*phsA*–R	GACGCAGTACGTACCTTTACTC
*tdcA*	Transcriptional regulator TdcA	*tdcA*–F	GATTCAGACCAGGAAAGCAGTA
		*tdcA*–R	AGCGATGTTGAGGCCTATTT
*hlyD*	Secretion protein HlyD	*hlyD*–F	GATAATGCGCAGGCGATAGA
		*hlyD*–R	GGTCGTCCTGACAAACCTTAC
*rfaC*	Lipopolysaccharide heptosyltransferase RfaC	*rfaC*–F	GCAACACCGGATTACGGATAAA
		*rfaC*–R	GCGGTAACACATCCACATAGTC
*lbpB*	Heat shock chaperone IbpB	*ibpB*–F	CCAGATCTTCCTGACGGAAAC
		*ibpB*–R	AGAGCTTCCCGCCCTATAA
*dedD*	Cell division protein DedD	*dedD*–F	CCGACGAGAATACGGGTTATTT
		*dedD*–R	CGCGGATAAAGTCAACGAGATA
*fumD*	Fumarate hydratase FumD	*fumD*–F	CTTAACACGCCCGCAATAAC
		*fumD*–R	AACTGTACCGCGAGATGTG
*pspB*	Envelope stress response protein PspB	*pspB*–F	TCGCTTTGCGACAGTTCT

## Data Availability

The data that support the findings of this study are openly available in Dryad at https://datadryad.org/stash/share/YQhXCuCr_OKgh5sJWVlU30FwWkEox42kv-LbNgB9Qd0.

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
