# Peer review of "Novel Insight into the Effects of CpxR on Salmonella enteritidis Cells during the Chlorhexidine Treatment and Non-Stressful Growing Conditions"

_ijms, 2021, doi:10.3390/ijms22168938_

Round 1
Reviewer 1 Report
The presented work investigates the influence of chlorhexidine to Salmonella physiology based on the proteome analysis. Authors showed significant differences in ∆cpxR mutant protein synthesis (up-regulation and down-regulation) when compared to the wild-type strain. In my opinion, the idea of research is in line with the current global problem of microbial resistance and understanding the molecular processes underlying this issue is a needed challenge. The deep analysis of this manuscript drove me to express my kind suggestions, which I believe be helpful for authors to evaluate their work once again before publication. Please be familiar with the comments below.
Abstract. A quote: “The development and spread of antimicrobial resistance is a significant global challenge”. What kind of resistance you mean? To antibiotics or biocides? Or both? Please clarify.
“the discovery of novel bacterial cellular targets and the critical pathways associated with antimicrobial resistance is needed”. Is your conclusion/work at all contains answer to that statement? I feel not. Please highlight in the manuscript (in Abstract also) which process connected to a presented broad protein analysis may confer resistance or susceptibility to CHX. I think that it run out. Bearing in mind the above sentence – discovery of novel cellular targets. And why did you use CHX out of hundreds known active substances? What did you expect? What was your hypothesis?
Introduction is based on quite old publications. For example, for 25 references, only one is from 2020. Please update citations. Moreover, in this section authors should also summarize the mechanism of action (MOA) of CHX against bacterial cells, underline that CHX is less effective to Gram-negative bacteria in comparison to Gram-positive. CHX MOA is totally omitted which seems to be not coincide with the title and idea of that work.
In the whole work… CHX cannot be described as a drug, it is an antiseptic. It is very important.
Results. 2.1. Are you sure that Figure 1 depicts growth curve? There is not any curve. Moreover, did you check anywhere, if differences in temperature of incubation influence mutants’ growth? Probably, the result will be similar, but it would be more convincing if you showed growth kinetics e.g. at 300C or 370C.
2.2. The title contains the word “minimal” but in methods is used “minimum”. Please standardize. Chemical structure should be involved in the section materials. By the way, in manuscript there is any suggestion about MOA against mutants considering the molecular structure of antimicrobial. I’m wondering why finding a precise concentration of CHX was so important? Please comment this. Normally, in this type of method, 2-fold dilution is used. So meticulous analysis makes me want to know of CHX MOA.
Fig. 3. Please add a MIC concentration in the bracket. Salmonella enterica should be written in italics.
Fig. 4. You are right, that we couldn’t see/expect significant disorders within cytosol and envelope topography, because of low (MIC) CHX concentration. I totally agree with that.
The following part (protein analysis) of presented work is impressive. Authors delivered a great amount of data and found interesting differences between strains, confirmed also with qRT-PCR. But finally, I would know authors’ opinions, which of them (I mean differences) they think, may be consider when we try to solve the problem of bacterial resistance. And simply, why ∆cpxR mutant was susceptible to CHX? Any idea? Main reason?
Discussion. I kindly recommend to take into account for instance a reference at the end of the sentence „It is well documented that the decrease of porin abundance in a prokaryotic cell is an adaptive response of these organisms to exposure to a wide spectrum of antimicrobials [23, 28-30]: Futoma-KoÅ‚och B, Bugla-PÅ‚oskoÅ„ska G, Dudek B, Dorotkiewicz-Jach A, Drulis-Kawa Z, Gamian A. Outer Membrane Proteins of Salmonella as Potential Markers of Resistance to Serum, Antibiotics and Biocides. Curr Med Chem. 2019;26(11):1960-1978. doi: 10.2174/0929867325666181031130851. PMID: 30378478. This paper contains detailed description of Salmonella resistance to antibiotics, biocides, cross-resistance in the background of changes within outer membrane proteins and efflux-systems.
Author Response
Reviewer # 1
Comments and Suggestions for Authors
The presented work investigates the influence of chlorhexidine to Salmonella physiology based on the proteome analysis. Authors showed significant differences in ∆cpxR mutant protein synthesis (up-regulation and down-regulation) when compared to the wild-type strain. In my opinion, the idea of research is in line with the current global problem of microbial resistance and understanding the molecular processes underlying this issue is a needed challenge. The deep analysis of this manuscript drove me to express my kind suggestions, which I believe be helpful for authors to evaluate their work once again before publication. Please be familiar with the comments below.
Response # 1. Thank you.
Abstract. A quote: “The development and spread of antimicrobial resistance is a significant global challenge”. What kind of resistance you mean? To antibiotics or biocides? Or both? Please clarify.
Response # 2. “Antimicrobial resistance” was referred to a wide spectrum of microbial resistance including both antibiotics and biocides. To clarify this point, the sentence has been revised and now it reads: “The development and spread of antibiotics and biocides resistance is a significant global challenge.”
“the discovery of novel bacterial cellular targets and the critical pathways associated with antimicrobial resistance is needed”. Is your conclusion/work at all contains answer to that statement? I feel not. Please highlight in the manuscript (in Abstract also) which process connected to a presented broad protein analysis may confer resistance or susceptibility to CHX. I think that it run out. Bearing in mind the above sentence – discovery of novel cellular targets. And why did you use CHX out of hundreds known active substances? What did you expect? What was your hypothesis?
Response # 3. The answer to the first question related to cellular targets can be the CpxR regulator itself. Please see our response # 10. Targeting this regulator alone will not provide satisfactory results. However, targeting this regulator in combination with the main antibiotic can give satisfactory results, even against multidrug-resistant isolates. The main advantage of targeting the CpxR regulator lies in the fact that this regulator affects not one but multiple cellular processes, including the major central metabolic processes and the envelope biogenesis itself.
Regarding the reason why we selected CHX, we included a statement in the “Introduction” that answers this question. Please look at the highlighted part: “In the current study, we investigated the role of both RpoE and CpxR, regulators on the susceptibility of growing Salmonella enterica serovar Enteritidis cells to the polycationic antimicrobial agent, chlorhexidine (CHX). The mechanism of action of CHX is associated with the interaction of positively charged CHX molecules with the negatively charged surface of a microbial cell. As a result of this interaction, the cell membrane tends to disorganize, leading to a “leaky cell” phenotype. To study the roles of major extracytoplasmic stress response regulators in the antimicrobial susceptibility, we used an antimicrobial agent that primarily acts on the cell membrane and subsequently triggers the extracytoplasmic stress response.”
In this study, we expected to learn the role of the CpxR and RpoE regulators in the physiology of Salmonella and more specifically to learn their roles in the antimicrobial susceptibility of this organism. There was no specific hypothesis. Actually, we carried this type of study to generate potentially interesting hypotheses.
Introduction is based on quite old publications. For example, for 25 references, only one is from 2020. Please update citations. Moreover, in this section authors should also summarize the mechanism of action (MOA) of CHX against bacterial cells, underline that CHX is less effective to Gram-negative bacteria in comparison to Gram-positive. CHX MOA is totally omitted which seems to be not coincide with the title and idea of that work.
Response # 4. Thanks for pointing out the references. It is important to emphasize that these 25 references can be divided into two distinct groups. While one small group (five publications) of references refer to the issue of antimicrobial resistance, the second (twenty publications) group of references are associated with the discoveries and physiological roles of CpxRA and RpoE. The first group of references has been updated accordingly, whereas the second group of references has not been updated. The reason for not updating the second group of references lies in the fact that these publications represent the research hallmarks in the biology or the CpxRA and RpoE regulators.
Regarding the second part of the reviewer’s comment, the mechanism of action of CHX has been included with the explanation of why CHX was selected as an antimicrobial agent. Now, this paragraph reads: “In the current study, we investigated the role of both RpoE and CpxR, regulators on the susceptibility of growing Salmonella enterica serovar Enteritidis cells to the polycationic antimicrobial agent, chlorhexidine (CHX). The mechanism of action of CHX is associated with the interaction of positively charged CHX molecules with the negatively charged surface of a microbial cell. As a result of this interaction, the cell membrane tends to disorganize, leading to a “leaky cell” phenotype. To study the roles of major extracytoplasmic stress response regulators in the antimicrobial susceptibility, we used an antimicrobial agent that primarily acts on the cell membrane and subsequently triggers the extracytoplasmic stress response.”
In the whole work… CHX cannot be described as a drug, it is an antiseptic. It is very important.
Response # 5. That is correct – CHX is categorized as an antiseptic. We used the term “drug” in a broad sense. To avoid any misinterpretation and further misleading of the readers, a term “drug” has been replaced with a term “antiseptic” and “CHX” throughout the manuscript.
Results. 2.1. Are you sure that Figure 1 depicts growth curve? There is not any curve. Moreover, did you check anywhere, if differences in temperature of incubation influence mutants’ growth? Probably, the result will be similar, but it would be more convincing if you showed growth kinetics e.g. at 300C or 370C.
Response # 6. Thank you for catching this mistake. The term “curves” has been replaced with a more appropriate term “rates”. Now, the text in this section reflects the data presented in Figure 1.
Yes, we agree that the incubation temperature of 37 °C would be in some sense more convincing compared to that of 22 °C used in this experiment. The main reason for the selection of 22 °C as an incubation temperature for this experiment lies in the fact that this temperature provides a longer growth time for this organism compared to that of 37 °C. For instance, using the incubation temperature of 22 °C, the growth (exponential) phase lasts approximately five to six hours, whereas the incubation temperature of 37 °C would result in a much shorter growth phase time of this organism. By extending the growth phase time, we intended to test this physiological trait (growth) over a longer period, subsequently increasing the sensitivity of this assay.
2.2. The title contains the word “minimal” but in methods is used “minimum”. Please standardize. Chemical structure should be involved in the section materials. By the way, in manuscript there is any suggestion about MOA against mutants considering the molecular structure of antimicrobial. I’m wondering why finding a precise concentration of CHX was so important? Please comment this. Normally, in this type of method, 2-fold dilution is used. So meticulous analysis makes me want to know of CHX MOA.
Response # 7. Thank you for catching this mistake. The word “minimal” has been replaced with the word “minimum” throughout the manuscript.
Regarding MOA against mutants and a question of “why finding a precise concentration of CHX was so important?” Generally, the mode of action of CHX against bacterial cells is associated with the disorganization of the cell membrane. The concentration of positively charged CHX molecules will determine the extent of bacterial cell damage. It ranges from bacteriostatic to bactericidal effect. In this study, the objective was to determine the roles of two major extracytoplasmic stress response regulators, RpoE and CpxR, during the treatment with an agent that triggers extracytoplasmic stress response. To do this, it was important to find out the minimum inhibitory concentration of CHX for the wild type. In other words, too high or too low concentrations of CHX could lead to no differences in susceptibilities to CHX between the wild type and its mutants.
Fig. 3. Please add a MIC concentration in the bracket. Salmonella enterica should be written in italics.
Response # 8. The missing information has been added. In addition, the name of the model organism has been corrected.
Fig. 4. You are right, that we couldn’t see/expect significant disorders within cytosol and envelope topography, because of low (MIC) CHX concentration. I totally agree with that.
Response # 9. Thank you.
The following part (protein analysis) of presented work is impressive. Authors delivered a great amount of data and found interesting differences between strains, confirmed also with qRT-PCR. But finally, I would know authors’ opinions, which of them (I mean differences) they think, may be consider when we try to solve the problem of bacterial resistance. And simply, why ∆cpxR mutant was susceptible to CHX? Any idea? Main reason?
Response # 10. Thank you for the comments. The problem of bacterial resistance is a very complex problem and in my opinion, requires a multifaceted approach on the global level. Regarding specifically the CpxR regulator and the susceptibility of Salmonella to CHX, I think that the main reason for the mutant susceptibility to CHX is associated with the significant and multiple roles of CpxR in the physiology of Salmonella. It was quite surprising to observe such a difference between the proteomes of the wild type and mutant under non-stressful (exponential growth phase). This study showed that the CpxR regulator plays an important role not only during extracytoplasmic stress but during non-stressful conditions. The present study showed that the dysfunctional CpxR regulators under non-stressful conditions affect the central metabolic pathways including the TCA cycle, electron transport and biogenesis of the cell envelope, which probably significantly contributes to an increased mutant’s susceptibility.
Discussion. I kindly recommend to take into account for instance a reference at the end of the sentence „It is well documented that the decrease of porin abundance in a prokaryotic cell is an adaptive response of these organisms to exposure to a wide spectrum of antimicrobials [23, 28-30]: Futoma-KoÅ‚och B, Bugla-PÅ‚oskoÅ„ska G, Dudek B, Dorotkiewicz-Jach A, Drulis-Kawa Z, Gamian A. Outer Membrane Proteins of Salmonella as Potential Markers of Resistance to Serum, Antibiotics and Biocides. Curr Med Chem. 2019;26(11):1960-1978. doi: 10.2174/0929867325666181031130851. PMID: 30378478. This paper contains detailed description of Salmonella resistance to antibiotics, biocides, cross-resistance in the background of changes within outer membrane proteins and efflux-systems.
Response # 11. Thank you for this reference. I did not come across this review article before, but it nicely fits with the “Discussion” of the manuscript. Therefore, Futoma-Kotoch et al. has been included as a reference in this manuscript.
Reviewer 2 Report
This manuscript presents a solid, logical treatment of its central issue and addresses the subject well. I have a few specific comments, however:
Line 138: "Red Lambda recombineering" should be "Lambda Red recombineering". Also, this is the first mention that this technique is being used at all and it may be worthwhile using a brief introduction of it.
Line 170-171: The terms "cell envelope" and "bacterial envelope" are not directly intuitive; it would be worthwhile to clarify which membranes are meant as in some contexts these terms are both used for the inner membrane, outer membrane, cell wall, or some fraction thereof.
From Line 406: The sharp difference in CHX from past studies warrants more explanation or a supplementary figure showing inhibition curve.
I would like to see more discussion of the specific genes whose expression is drastically altered, in addition to the discussion of categories. Why are these genes so significant in response?
Author Response
Reviewer # 2
Comments and Suggestions for Authors
This manuscript presents a solid, logical treatment of its central issue and addresses the subject well. I have a few specific comments, however:
Response # 12. Thank you.
Line 138: "Red Lambda recombineering" should be "Lambda Red recombineering". Also, this is the first mention that this technique is being used at all and it may be worthwhile using a brief introduction of it.
Response # 13. Thank you for catching this mistake. It has been corrected and brief info regarding this technique has been included.
Line 170-171: The terms "cell envelope" and "bacterial envelope" are not directly intuitive; it would be worthwhile to clarify which membranes are meant as in some contexts these terms are both used for the inner membrane, outer membrane, cell wall, or some fraction thereof.
Response # 14. Agreed. These terms can be confusing sometimes, as it seems they are not quite standardized. To avoid the confusion of the reader, the sentence has been revised accordingly. Now, it reads: “This indicates that the initial lethality of the ΔcpxR mutant during the drug treatment was not associated with major structural perturbation of the bacterial envelope, but rather via less invasive changes of the cellular envelope, primarily outer membrane and secondary periplasm and inner (plasma) membrane.”
From Line 406: The sharp difference in CHX from past studies warrants more explanation or a supplementary figure showing inhibition curve.
Response # 15. Agreed. The paragraph has been revised and now it reads: “Tattawasart et al. [36], using a similar approach, found that the CHX treatment caused disorganization of the outer membrane, a significant loss of cytoplasmic material and finally lysis of Gram-negative organism, Pseudomonas stutzeri. It is worth noting that Tattawasart et al. [36] used 100 mg/L concentration of CHX, whereas in our study we used 3 mg/L, as this concentration proved to be a MIC for our model organism. It is important to emphasize that the objectives of these two studies were different. While Tattawasart et al. [36] aimed to reveal the effect of CHX on the cellular structure, we aimed to reveal the role of the CpxR regulator in the susceptibility of Salmonella to CHX. Importantly, our TEM analyses showed that the initial lethality of the cpxR mutant treated with MIC of CHX was not associated with the loss of cellular turgor or major membrane perturbation but rather with less invasive mechanisms.”
I would like to see more discussion of the specific genes whose expression is drastically altered, in addition to the discussion of categories. Why are these genes so significant in response?
Response # 16. Agreed. This is an interesting point. Therefore, we focused on the most significantly altered proteins and included the following statement:
“In addition, it is worthy of mentioning that certain proteins, including RelB, ClpB and ribonuclease E (rne), showed an outstanding increase in abundance compared to other altered proteins in their groups, DNA repair, chaperone and folding catalysts and RNA degradation, respectively. It is important to emphasize that one of the hallmarks of the stress response chaperones, ClpB, showed an outstanding increase in abundance, indicating an increased level of aggregated proteins in the ∆cpxR background.”
Reviewer 3 Report
The manuscript of S. Vidovic et al. titled “Novel insight into the effects of CpxR on Salmonella Enteritidis cells during the chlorhexidine treatment and non-stressful growing conditions” devoted to the effect of cpxR gene deletion on the functions of S. Enteritidis during the chlorhexidine treatment by using the proteomics assay. The study is well designed and I didn’t find any issues.
The next are some minor issues and comments that can be easily resolved:
Lines 85 and 106: You can shorten to S. Enteritidis
Figure 2: Add significance bars and stars
Figure 5 C and E: what means stars?
Line 325: omit quotation marks for SPI-1
Line 571: Change PMC… to ref. 46
Line 573: it is not clear which reference sequence was used. Add GenBank acc. no. and strain name.
Line 635: doi link is not working.
Author Response
Reviewer # 3
Comments and Suggestions for Authors
The manuscript of S. Vidovic et al. titled “Novel insight into the effects of CpxR on Salmonella Enteritidis cells during the chlorhexidine treatment and non-stressful growing conditions” devoted to the effect of cpxR gene deletion on the functions of S. Enteritidis during the chlorhexidine treatment by using the proteomics assay. The study is well designed and I didn’t find any issues.
The next are some minor issues and comments that can be easily resolved:
Response # 17. Thank you.
Lines 85 and 106: You can shorten to S. Enteritidis
Response # 18. These changes have been introduced.
Figure 2: Add significance bars and stars
Response # 19. Please note that only one biological replication was carried out for the experiment presented in Figure 2 B. For that reason, no bars could be included. However, bars are included in Figure 2 C.
Figure 5 C and E: what means stars?
Response # 20. Thank you for catching this. Now legends of Figure 5 C and E contain the names of indicated proteins. The reason for indicating these proteins is that they do not have a standard protein nomenclature.
Line 325: omit quotation marks for SPI-1
Response # 21. It has been omitted.
Line 571: Change PMC… to ref. 46
Response # 22. It has been changed.
Line 573: it is not clear which reference sequence was used. Add GenBank acc. no. and strain name.
Response # 23. Agreed. We included the name of the reference Salmonella strain and its accession number.
Line 635: doi link is not working.
Response # 24. Thank you for letting us know. The new, workable, link has been added.